# Randomness as a driver of inactivity in social groups

**Abel Bernadou, Raphaël Jeanson** [ID] *

Centre de Recherches sur la Cognition Animale, Centre de Biologie Intégrative, Université de Toulouse, CNRS, UPS, France

* raphael.jeanson@univ-tlse3.fr

## Abstract

Social insects, such as ants and bees, are known for their highly efficient and structured colonies. Division of labour, in which each member of the colony has a specific role, is considered to be one major driver of their ecological success. However, empirical evidence has accumulated showing that many workers, sometimes more than half, remain idle in insect societies. Several hypotheses have been put forward to explain these patterns, but none provides a consensual explanation. Task specialisation exploits inter-individual variations, which are mainly influenced by genetic factors beyond the control of the colony. As a result, individuals may also differ in the efficiency with which they perform tasks. In this context, we aimed to test the hypothesis that colonies generate a large number of individuals in order to recruit only the most efficient to perform tasks, at the cost of producing and maintaining a fraction of workers that remain inactive. We developed a model to explore the conditions under which variations in the scaling of workers' production and maintenance costs, along with activity costs, allow colonies to sustain a fraction of inactive workers. We sampled individual performances according to different random distributions in order to simulate the variability associated with worker efficiency. Our results show that the inactivity of part of the workforce can be beneficial for a wide range of parameters if it allows colonies to select the most efficient workers. In decentralised systems such as insect societies, we suggest that inactivity is a by-product of the random processes associated with the generation of individuals whose performance levels cannot be controlled.

## Author summary

The presence of many inactive workers in social insect colonies is puzzling, and so far, there are no clear explanations for this phenomenon. Behavioural variation within the workforce plays a key role in the division of labour. The mechanisms underpinning this diversity cannot be, however, fully controlled, and may result in the production of inefficient workers. Using a modelling approach, we showed that colonies benefit from the production of a large workforce if this allows them to recruit the most competent workers, despite the costs associated with maintaining the remaining fraction as inactive. We propose that the presence of inactive workers has no primary functional role in colonies and

**Data Availability Statement:** The Java code to run simulations is provided in the Supplementary Information.

**Funding:** This work was funded by the Centre National de la Recherche Scientifique, Université Toulouse III - Paul Sabatier, and the Agence

Nationale de la Recherche (ANR-21-CE37-0001 to RJ and ANR- 22-CPJ2-0133-01 to AB). The funders had no role in study design, data collection and analysis, decision to publish, or preparation of the manuscript.

**Competing interests:** The authors have declared that no competing interests exist.

is simply a by-product of a random process aimed at generating a competent workforce. Our study, therefore, can explain the existence of inactivity in social groups across taxa.

## Introduction

Colonies of eusocial insects like ants, corbiculate bees, and termites are renowned for their bustling activity. However, several studies reveal a surprising and yet common feature of these societies: a significant portion of the workers, sometimes more than half of them, remains inactive for extended periods of time [1,2]. Why would colonies invest resources in producing a large workforce if a substantial number of individuals remain inactive? Both theoretical and experimental work have explored the potential adaptive benefits of inactivity [3–7]. A frequently invoked hypothesis proposes that these inactive individuals act as a reserve workforce, readily available to respond to surges in colony needs [4,8]. This assumption is based on the fact that individuals with high response thresholds, previously inactive, are recruited when task demands increase or when workers previously engaged in those tasks are lost [3]. However, empirical support for this hypothesis is mixed. For example, in *Temnothorax rugatulus* ant colonies, removing the most active workers leads to the recruitment of previously inactive workers to compensate for this loss [4]. However, in bumble bee colonies, removing the most active individuals before an increased need for defence or thermoregulation did not activate idle workers [9,10]. Instead, already active bees increased their activity to compensate. Even if inactive workers can be recruited when the colony's needs increase in certain circumstances, we can still question the original cause of inactivity. So far, no comprehensive hypothesis has been put forward to explain the widespread, and seemingly universal, occurrence of inactivity in social insect colonies. Identifying a common origin of inactivity across different taxa implies finding a mechanism shared by social insects that is related to the division of labour, with individual variations being key [11,12]. Here, we propose that inactivity could be a by-product of a process that selectively favours the best-performing workers when individual performance is determined randomly.

The interplay between genetic and non-genetic factors shapes the behavioural diversity of workers. Environmental factors like food and temperature experienced by the brood during development can be partially controlled by nurses. For example, exposure of ant brood to different temperatures can influence worker sensitivity to thermal cues in adulthood [13]. On the other hand, the genetic background of workers, produced from fertilized eggs laid by the queen, remains beyond any control. These very mechanisms that foster a diverse workforce may also result in colonies producing workers with unpredictable efficiency. If worker efficiency varies, a potential optimal strategy from a colony perspective might involve producing a large number of individuals to increase the likelihood of obtaining highly efficient ones. This ensures that the most competent workers complete tasks efficiently, even if this entails costs by keeping others in an inactive state. We aim to test this hypothesis from a theoretical perspective, considering the costs and benefits associated with the presence of inactive individuals within colonies.

Two types of costs are usually distinguished: production costs arise from the renewal of the workforce, while maintenance costs are incurred for workers' operation [14]. The later differ depending on activity level, with inactive workers incurring lower costs compared to those actively engaged in task completion. In the leaf-cutter ant *Atta sexdens*, for example, ants engaged in digging expend 15% more energy than their inactive nestmates [15]. Active workers can also vary in their performance when engaged in their tasks, with some individuals

being outperforming others. Foraging success and defensive behaviours, for instance, can vary significantly among workers in social insects [16–20].

In this context, we introduce a simple model to test the influence on colony performance of the ratio between maintenance costs and production costs (hereafter, coefficient δ), as well as of the ratio between maintenance costs of inactive individuals and those of active individuals (hereafter, coefficient $\beta$), when workers' performances are drawn from different random distributions.

## Results and discussion

For each of five random distributions and different combinations of δ and $\beta$ values, we determined which optimal fraction of active workers, chosen among the individuals with the highest performance and leaving the other workers inactive, maximized colony efficiency (Fig 1). The notion of individual performance here indicates that workers vary in their sensitivity to task-associated stimuli (i.e., differences in response thresholds [11]) or in their effectiveness in completing tasks. The selection of the most efficient workers can be understood by the competition between workers, where only the most 'motivated' engage in tasks [21]. Our model shows that it can be beneficial for colonies to leave low-performing workers inactive when their maintenance costs are high relative to their production costs (i.e., high δ), if this allows the best-performing individuals to be recruited. Such strategy is also advantageous if the maintenance costs of active workers are higher than those of inactive individuals (i.e., low $\beta$). In contrast, when the maintenance costs of active and inactive workers are comparable (i.e., high $\beta$), colonies should minimise the fraction of inactive individuals. Under these conditions, maintaining inactive workers would only produce marginal savings, so it is worthwhile for the colonies to recruit all individuals, including those with low individual performance. Evaluating efficiency by the difference between performance and costs, rather than the ratio, yielded similar results (S1 Fig), confirming the robustness of our conclusions regardless of the calculation method. Interestingly, the different random distributions we explored gave rise to similar qualitative patterns, demonstrating the generality of our findings. The normal, uniform and bimodal distributions used in our simulations have the same mean ($\mu = 0.5$) for individual performance but differed in their standard deviations (SD = 0.19, SD = 0.29, SD = 0.39, respectively). Our results showed that increasing variance amplified the range of parameter values ($\beta$, δ) where maintaining a pool of inactive workers is most profitable for colonies (Fig 1).

When the distribution of individual performances is left-skewed, with the majority of group members being high performers, colonies maximize efficiency by keeping all workers active. In situations of intraspecific competition, colonies might aim to direct worker production towards the most efficient individuals in order to out-compete rival groups. However, since colonies of the same species share the same mechanisms and constraints in worker production, they are equally influenced by chance in determining individual quality. Thus, no colony should be able to dominate intraspecific competition because of its ability to control the distribution of workers.

Several studies have documented that individual worker performance often follows a right-skewed distribution [18–20]. Our simulations show that the highest fraction of inactive individuals maximizing colony efficiency was obtained with such right-skewed distribution (Fig 1). Empirical data also suggests that, in ant colonies, maintenance costs typically exceed production costs (i.e., δ>1) [14]. This finding aligns with the value found in our simulations, where maintaining an inactive fraction of workers becomes advantageous. The higher the ratio of maintenance costs to production costs, the greater the benefit of keeping a large proportion of workers inactive. Refining our model to include different costs associated with morphological

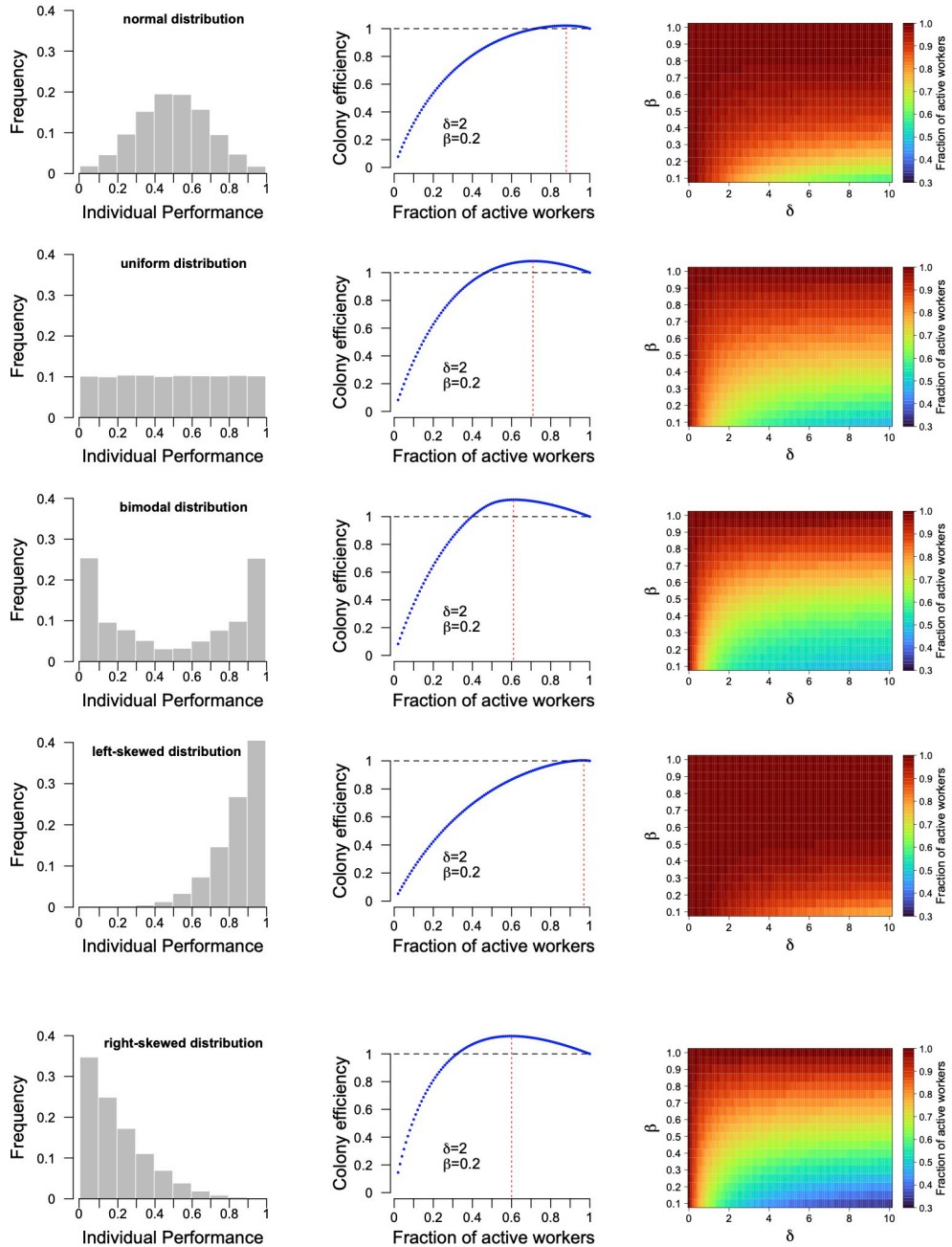

**Fig 1. Colony efficiency as a function of the proportion of active workers in colonies of 500 individuals.** Left: Histogram of individual task performances for five different random distributions. Middle: Colony efficiency as a function of the fraction of active workers for values of δ = 2 and $\beta$ = 0.2 (arbitrary choice of values to illustrate methods). δ corresponds to the ratio of maintenance costs to production costs and $\beta$ to the ratio of maintenance costs for inactive to active individuals. For each fraction of active workers, colony efficiency is normalized by the value obtained when all individuals are active. The red dotted line indicates the fraction of active individuals maximizing colony efficiency for this combination of δ and $\beta$ values. Right: Heat maps of the fraction of individuals that need to be active to maximize colony efficiency for different combinations of δ and $\beta$. For results with group size of 50 and 5,000 see S1 Fig.

differences, variations in performance over an individual's lifetime or an influence of group size on social dynamics would not change our main conclusions. Overall, the best strategy for colonies with inefficient workers is to keep them inactive, unless they are cheap to maintain and their cost of activity is low.

Assuming that variations in colony size have no influence at the individual level, our analysis also showed that changes in the number of workers had no impact on the patterns observed (S2 Fig). However, groups cannot expand indefinitely, as this entails additional costs, such as an increased risk of infection or greater competition [22]. In multi-level societies, fission-fusion dynamics can mitigate the costs associated with belonging to a large group by splitting into smaller units [23], but social insect colonies generally remain cohesive (except when colonies are swarming to reproduce). Further analysis should assess how the societies with different social organisations and levels of cohesion deal with the presence of inactive individuals.

Our conclusions are based on the assumption that colonies have no control over workers' performance. This hypothesis is consistent with the occurrence of multiple mating in several species of social insects, such as in honey bees where queens can mate with up to 30 males, which is a means of increasing the breadth of distributions where workers are sampled and improving colony homeostasis [24]. However, to a certain extent, societies can develop strategies to minimise the impact of chance on the behavioural trajectory of individuals. For instance, around 30% of ant genera exhibit polymorphism with workers from distinct physical sub-castes tending to engage in different tasks [25,26]. These variations in worker body size are primarily attributed to epigenetic factors, such as larval nutrition or exposure to abiotic factors that are under the control of the workers [27]. In the ant *Pheidole pallidula*, for example, foragers exposed to competitors during foraging bouts direct brood development towards the production of soldiers to improve colony defence [28]. From our perspective, the evolution of polymorphism can be viewed as a mechanism for minimizing the role of chance in determining the behavioural destiny of the workforce. However, this reduces but does not eliminate the contribution of randomness, as stochasticity is likely to shape individual behaviour within each subcaste. Such polymorphic species would be ideal systems for testing our model. In these species, the ratio of maintenance to production costs (i.e., $\delta$) is higher for major workers than for minors [29] and major workers tend to exhibit more inactivity than minor workers [30]. However, we currently lack data on how maintenance costs differ between active and inactive states within each caste (value of $\beta$), which complicates predictions about how activity levels should vary between castes according to our model.

The presence of inactive workers does not necessarily jeopardize colonies because the remaining active individuals are sufficient to cover all needs. In ants, for instance, the energy costs expended by a forager on a foraging trip are far outweighed by the energy benefits gained from the retrieved food [31,32]. Even if there are variations between species, the energy returns from successful foraging trips can exceed the metabolic costs of its foraging trips by several orders of magnitude [33]. Therefore, a fraction of active individuals can readily cover the expenses of the colonies, making inactive individuals less of a burden. However, it is important to note that this strategy might not work in other collective systems. The inactivity of a portion of the group could be detrimental if the remaining active individuals cannot fulfil all requirements or if inactive elements directly hinder overall efficiency. For instance, in a sports team, inactive players can ruin collective performance, such as in tug-of-war or rowing, where success depends on everyone's effort.

Our results can explain the presence of inactive workers within social systems and shed light on why no adaptive hypothesis has been successful in providing a compelling justification for their widespread existence. One might wonder why colonies do not eliminate inactive workers to reduce their costs, even if minimal. In our opinion, colonies lack the ability to

evaluate individuals' performance on collective outcome. This might also be true for other systems lacking supervision or control mechanisms to assess individuals' performance. In contrast, more centralized systems such as human corporations implement recruitment procedures based on an evaluation of individual skills and competences, thereby reducing the influence of random processes in selection. Our conclusion that inactive individuals can be maintained within colonies because there is no control over their performance echoes one explanation introduced to account for redundancy in collectives [34]. Redundancy—where groups maintain more members than are strictly needed for task completion—has been observed in a range of biological systems, from cellular assemblies to animal societies [35–37]. The challenge is to explain how groups avoid the invasion of defectors when individuals pay high costs but contribute minimally to the outcome. One hypothesis suggests that cooperative behaviours are still possible when individuals are unaware of others' strategies [34]. The inability of group members to evaluate the efforts or contribution of others may therefore play an important, but often overlooked, role in shaping workload distribution in collective systems.

In conclusion, our study shows that the presence of inactive individuals is not necessarily detrimental to the functioning of society. The strength of our simple model lies in introducing a new hypothesis: that a single mechanism can explain inactivity in social groups, something previous hypotheses have not successfully achieved. The presence of inactive individuals could then be co-opted at later stages to fulfil functions that were not the primary cause of their existence. For instance, these individuals could act as a reserve workforce that can be recruited to cover certain additional needs or to serve as living reservoirs to store food [1].

Overall, our results should encourage us to consider that any behaviour should not be regarded as adaptive, but that it may be the result of an uncontrollable random process, a dimension frequently underestimated in the study of social systems.

## Methods

We simulated colonies of constant size, taking into account the production and maintenance costs, and performance at the colony level.

### Colony costs

The costs of production of a worker ($\text{Costs}_{\text{Worker}_{\text{Production}}}$) correspond to the energy invested in the making of a new individual. The costs of maintenance of a worker ($\text{Costs}_{\text{Worker}_{\text{Maintenance}}}$) comprise the energy required to fuel the activity of an adult, as well as the indirect costs associated with maintaining an active workforce, such as the risk of disease transmission by foragers or predation risk on outside workers. We suppose that all workers have the same body size (*i.e.*, no polymorphism), so production costs are equal for all individuals. The production costs borne by a colony are:

$$\text{Costs}_{\text{Colony}_{\text{Production}}} = \text{Costs}_{\text{Worker}_{\text{Production}}}\left(N_{\text{active}} + N_{\text{inactive}}\right)$$

Maintenance costs can vary based on an individual's level of activity, with inactive workers incurring lower costs than active ones. To take this into account, we introduce the coefficient $\beta$, where a value of $\beta$ closer to zero indicates lower maintenance costs for inactive individuals compared to active ones. Maintenance costs paid by the colony are given by the sum of maintenance costs of all workers:

$$\text{Costs}_{\text{Colony}_{\text{Maintenance}}} = \text{Costs}_{\text{Worker}_{\text{Maintenance}}}\left(N_{\text{active}} + \beta N_{\text{inactive}}\right)$$

with $N_{\text{active}}$ and $N_{\text{inactive}}$, the number of active and inactive workers.

The total colony costs are:

$$\text{Costs}_{\text{Colony}} = \text{Costs}_{\text{Colony}_{\text{Production}}} + \text{Costs}_{\text{Colony}_{\text{Maintenance}}}$$

We introduce the coefficient δ, which represents the ratio between the maintenance costs of a worker over its production costs. Therefore, the colony costs are given by:

$$\text{Costs}_{\text{Colony}} = \text{Costs}_{\text{Worker}_{\text{Production}}} (N_{\text{active}} + N_{\text{inactive}}) + \delta \text{Costs}_{\text{Worker}_{\text{Production}}} (N_{\text{active}} + \beta N_{\text{inactive}})$$

## Individual and colony performances

By definition, only active workers contribute to the functioning of the colony, but they can have different levels of performance. Here, the term "performance" is used broadly to indicate that workers differ in their efficiency at task completion, either because they are differently sensitive to task-associated stimuli (i.e., differences in response thresholds [11]) and/or they are more or less effective in performing the task. We examined five scenarios in which individual performances were sampled from five distinct types of random distributions in the interval [0, 1]: a truncated normal distribution (with mean = 0.5, SD = 0.2), a truncated bimodal distribution (mean$_1$ = 0.1, SD$_1$ = 0.2; mean$_2$ = 0.9, SD$_2$ = 0.2) a uniform distribution, a right-skewed distribution and a left-skewed distribution (Fig 1). For the left-skewed distribution, random numbers were calculated as $u^{(1/skewness)}$, and for the right-skewed distribution, as $1 - u^{(1/skewness)}$ ($u$: random numbers drawn uniformly between 0 and 1, *skewness* = 4). The colony's performance is given by the sum of the individual performance of the active workers.

$$\text{Performance}_{\text{Colony}} = \sum_{i=1}^{i=N_{\text{active}}} (\text{Performance}_i)$$

Efficiency$_{\text{Colony}}$ is given by dividing Performance$_{\text{Colony}}$ by Costs$_{\text{Colony}}$. We also examined whether measuring Efficiency$_{\text{Colony}}$ by subtracting Costs$_{\text{Colony}}$ from Performance$_{\text{Colony}}$ yielded similar conclusions (S2 Fig).

## Simulations

We simulated colonies of 50, 500 and 5000 individuals over a period equivalent to the lifetime of workers (this means that if we assume a constant mortality rate, the renewal of all workers takes place over a period equal to the lifetime of a worker). We assume that individual characteristics remain consistent throughout their lifetimes, thus excluding any maturation or aging effects. We set the production costs of a worker at 100 arbitrary units. For each random distributions, we calculated for different combinations of $\beta$ (from 0.1 to 1 in increments of 0.05) and δ (from 0 to 10 in increments of 0.1) and different numbers of active workers (from 10 to 50, 500 or 5,000 in increments of 5). We assume that active workers have equal maintenance costs, irrespective of their performance. In each simulation, the individual performance of each worker was randomly drawn. The active individuals selected were those with the highest individual performance (for example, if 50 individuals were selected as active, we evaluated the colony's performance by aggregating the individual performances of the 50 best-performing workers). For each combination of $\beta$ and δ, we normalized the values of colony efficiency obtained for each fraction of active workers by the value obtained when all workers were active (*i.e.*, highest possible colony performance). Next, we determined the value of the fraction of active workers maximising Efficiency$_{\text{Colony}}$. For each combination of δ and $\beta$ and each fraction of active individuals, 1 000 simulations were run.

## Supporting information

**S1 Text. Java code to run simulations.**
(RTF)

**S1 Fig. Alternative methods to compute efficiency.** Colony efficiency as a function of the proportion of active workers. Left: Histogram of individual task performance for five different random distributions. Right: colony efficiency as a function of the proportion of active workers for values of δ = 2 and β = 0.2. Blue lines: efficiency measured as the ratio of performance to cost (as in Fig 1); red lines: efficiency measured as the difference between performance and cost. See Fig 1 for details.
(TIF)

**S2 Fig. Influence of colony size.** Colony efficiency as a function of the proportion of active workers for different colony size (50, 500, 5000 workers). Left: Histogram of individual task performances for five different random distributions. One thousand replicates were simulated for colonies of 50 and 500 workers and 100 replicates for colonies of 5000 workers. See Fig 1 for details.
(TIF)

## Author Contributions

**Conceptualization:** Abel Bernadou, Raphaël Jeanson.

**Data curation:** Raphaël Jeanson.

**Formal analysis:** Abel Bernadou, Raphaël Jeanson.

**Funding acquisition:** Abel Bernadou, Raphaël Jeanson.

**Investigation:** Abel Bernadou, Raphaël Jeanson.

**Methodology:** Abel Bernadou, Raphaël Jeanson.

**Validation:** Abel Bernadou, Raphaël Jeanson.

**Visualization:** Abel Bernadou, Raphaël Jeanson.

**Writing – original draft:** Abel Bernadou, Raphaël Jeanson.

**Writing – review & editing:** Abel Bernadou, Raphaël Jeanson.

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
