## [Decision Letter · Decision Letter 0]

20 Sep 2024

Dear Dr. Jeanson,

Thank you very much for submitting your manuscript "Randomness as a driver of inactivity in social groups" for consideration at PLOS Computational Biology.

As with all papers reviewed by the journal, your manuscript was reviewed by members of the editorial board and by several independent reviewers. In light of the reviews (below this email), we would like to invite the resubmission of a significantly-revised version that takes into account the reviewers' comments.

Both reviewers have constructively suggested revisions in terms of further explaining the model (and analysis) and including additional discussion around the current state of the literature, with specific examples. I encourage you to consider their comments.

We cannot make any decision about publication until we have seen the revised manuscript and your response to the reviewers' comments. Your revised manuscript is also likely to be sent to reviewers for further evaluation.

Sincerely,

Stuart Johnston

Academic Editor

PLOS Computational Biology

Tobias Bollenbach

Section Editor

PLOS Computational Biology

Both reviewers have constructively suggested revisions in terms of further explaining the model (and analysis) and including additional discussion around the current state of the literature, with specific examples. I encourage you to consider their comments.

Reviewer's Responses to Questions

**Comments to the Authors:**

Reviewer #1: the review is uploaded

Reviewer #2: In this manuscript, authors develop a model to examine worker inactivity in social insects like ants and bees. They support the idea that colonies generate many individuals to recruit the most efficient workers, despite the cost of maintaining inactive ones. Inactivity is suggested as a by-product of random processes “associated with the generation of individuals whose performance levels cannot be controlled”. The manuscript is well written, and the model comprehensively presented. I am not a theoretician, and therefore I can’t evaluate the technical details of the model. Instead, I have focused my review on the framing of the manuscript and on its broader contributions to the field. I am very positive about it but I think it needs quite a bit of work on presenting the current state of knowledge and the existing gaps. I hope that is still a helpful and constructive review.

The literature on the concept of redundancy in eusocial insect societies is quite rich, see for example Pedroso (2021). This paper is not discussed in the manuscript, despite it explores whether “redundancy can evolve as a by-product of selection when group members have to opt whether to cooperate or not while being blind to the strategies of others”. Pedroso’s work views the concept of redundancy through the lens of cooperation, and this perspective is overlooked in the current manuscript. Broadly, empirical tests and theoretical models have been employed to explore why there are inactive workers in animal societies. However, I believe that the manuscript doesn’t discuss the existing knowledge and developing hypotheses on the topic. This lack of background information doesn’t help the reader understand the new perspective that the manuscript contributes, where it agrees, and where it disagrees with the literature. I think that to improve this authors need to do literature searches, study their findings and re-write the introduction and discussion.

Another point I have is that beyond the production and maintenance costs of active and inactive workers, maintaining a large group size can also imply a broad range of extra costs (Krause & Ruxton, 2002), such as an increased risk for the spread of diseases or even higher inequality in the division of labour among group members (Rotics & Clutton-Brock, 2021). Authors should discuss additional costs associated with group size, as well as types of social organization, such as multilevel societies, which buffer those costs by allowing groups to associate when needed without maintaining a large group size at all times [see Camerlenghi & Papageorgiou (2024)].

L39 Why “mixed” support? Take space and time to explain the differences in these studies. I think this paragraph needs to be unpacked, expanded and explain what’s the new perspective that the current manuscript contributes, where it agrees and where it disagrees with the literature.

L50-51 Why to assume that a larger number of workers implies higher variation? This has to be supported with references or simulations.

L85 “This type” is unclear.

L100 This (Grüter et al., 2012) would be an interesting study to discuss here. I think the manuscript needs enrichment in terms of biological examples and animal species stories. PLOS Computational Biology is read by a wide audience that isn’t necessary familiar with the theory inspired by eusocial insects, therefore expanding the introduction and discussion in this direction would make the text more engaging and relevant for the readers of the journal.

L125-135 The flow in this part isn’t great and it repeats many points of the introduction. I think what would be very interesting to discuss is how the findings of this manuscript link/complement other models and empirical studies on the concept of redundancy in animal societies. Are there mutually exclusive hypotheses or the manuscript provides an additional explanation?

L136 Give ecological examples on what these functions could be.

L135-139 This feels like repeating what was said before in the manuscript.

L150 also the space they occupy in a shelter?

L196-197 I would discuss this assumption further because in eusocial insect societies with high levels of task specialisation and division of labour, body size can vary greatly and therefore maintenance costs across individuals may also vary.

L189 Why 500? What could be the effect of group size on all this? Discussion is missing.

References

Camerlenghi, E., & Papageorgiou, D. (2024). Multilevel societies: different tasks at different social levels. EcoEvoRxiv. https://doi.org/10.32942/X27C90

Grüter, C., Menezes, C., Imperatriz-Fonseca, V. L., & Ratnieks, F. L. W. (2012). A morphologically specialized soldier caste improves colony defense in a neotropical eusocial bee. Proceedings of the National Academy of Sciences, 109(4), 1182–1186. https://doi.org/10.1073/pnas.1113398109

Krause, J., & Ruxton, G. (2002). Living in groups. Oxford University Press.

Pedroso, M. (2021). Blind Cooperation: The Evolution of Redundancy via Ignorance. The British Journal for the Philosophy of Science, 72(3), 701–715. https://doi.org/10.1093/bjps/axz022

Rotics, S., & Clutton-Brock, T. (2021). Group size increases inequality in cooperative behaviour. Proceedings of the Royal Society B: Biological Sciences, 288(1945). https://doi.org/10.1098/rspb.2020.2104

**Have the authors made all data and (if applicable) computational code underlying the findings in their manuscript fully available?**

Reviewer #1: Yes

Reviewer #2: Yes

PLOS authors have the option to publish the peer review history of their article (what does this mean?). If published, this will include your full peer review and any attached files.

Reviewer #1: No

Reviewer #2: No
---

## [Decision Letter · Decision Letter 1]

15 Nov 2024

PCOMPBIOL-D-24-00988R1Randomness as a driver of inactivity in social groupsPLOS Computational Biology Dear Dr. Jeanson, Thank you for submitting your manuscript to PLOS Computational Biology. After careful consideration, we feel that it has merit but does not fully meet PLOS Computational Biology's publication criteria as it currently stands. Therefore, we invite you to submit a revised version of the manuscript that addresses the points raised during the review process. Please submit your revised manuscript within 30 days Jan 15 2025 11:59PM. If you will need more time than this to complete your revisions, please reply to this message or contact the journal office at ploscompbiol@plos.org. Please include the following items when submitting your revised manuscript:*
A rebuttal letter that responds to each point raised by the editor and reviewer(s). You should upload this letter as a separate file labeled 'Response to Reviewers'. This file does not need to include responses to formatting updates and technical items listed in the 'Journal Requirements' section below.*
A marked-up copy of your manuscript that highlights changes made to the original version. You should upload this as a separate file labeled 'Revised Manuscript with Track Changes'.*
An unmarked version of your revised paper without tracked changes. You should upload this as a separate file labeled 'Manuscript'. If you would like to make changes to your financial disclosure, competing interests statement, or data availability statement, please make these updates within the submission form at the time of resubmission. Guidelines for resubmitting your figure files are available below the reviewer comments at the end of this letter. We look forward to receiving your revised manuscript. Kind regards, Stuart JohnstonAcademic EditorPLOS Computational Biology Tobias BollenbachSection EditorPLOS Computational Biology

Feilim Mac Gabhann

Editor-in-Chief

PLOS Computational Biology

Jason Papin

Editor-in-Chief

PLOS Computational Biology

 **Additional Editor Comments (if provided):** Thank you for your constructive responses to the Reviewers. I agree with Reviewer 2 that discussion on potential additional costs associated with larger colony sizes would add to the appeal of the manuscript and I encourage you to consider including this.  **Journal Requirements:****Reviewers' comments:** Reviewer's Responses to Questions

**Comments to the Authors:**

Reviewer #1: I highly recommend this article for publication. The authors have largely taken my suggestions into account. In particular, they have included new results/analyses (concerning the influence of the size of the colony, criteria of efficiency).

I would like congratulate the authors for their stimulating contribution. Moreover I suggest that they continue to exploit this model in future theoretical and experimental works.

Reviewer #2: Review on the revised version of “Randomness as a driver of inactivity in social groups”

The authors have thoroughly revised their manuscript, but I think that one of my main comments was not addressed, with insufficient reasoning:

Reviewer 2, previous round: Another point I have is that beyond the production and maintenance costs of active and inactive workers, maintaining a large group size can also imply a broad range of extra costs (Krause & Ruxton, 2002), such as an increased risk for the spread of diseases or even higher inequality in the division of labour among group members (Rotics & Clutton-Brock, 2021). Authors should discuss additional costs associated with group size, as well as types of social organization, such as multilevel societies, which buffer those costs by allowing groups to associate when needed without maintaining a large group size at all times [see Camerlenghi & Papageorgiou (2024)].

Authors, previous round: As we do not discuss the link between group size and the level of inactivity, as this would be beyond the scope of this paper, we feel it is not necessary to comment on the costs associated with increasing colony size.

Reviewer 2, current round: Especially now that authors have run their simulations on three different group sizes, this discussion is missing even more than before. Inactive workers imply a larger group size (B) than that of a group with the same number of active workers but no inactive ones (A). In group B the cost of maintenance of inactive workers is larger than A, but there are multiple other costs that are not discussed in this manuscript, restrict group size in nature and are of critical importance, even if they aren’t examined through simulations here. In the previous round I mentioned only a few examples, but the list is longer. The example of the multilevel societies is also relevant as it shows how extra group members can be used when it’s needed, showing thus an alternative to the evolution of inactivity. I believe that a paragraph on these matters can be added in the discussion and would make the manuscript even more appealing.

Below I list line numbers on the document with track changes. I found quite a few typos and I believe that the manuscript should be read very carefully before resubmission.

L2 “organised” is quite vague, as most animal societies present some kind of organisation. For definition of social organisation see Kappeler 2019 in Behavioural Ecology.

L22 “there are no clear explanations” or “there is no clear explanation”

L79 “than” should be deleted.

L184 “might also be true” instead of “might be also true”

L192 I am not sure about reference 33. Birch is a philosopher, and this paper is not a systematic review but a philosophical perspective. There are plenty of empirical studies on redundancy in biological systems that could be cited.

**Have the authors made all data and (if applicable) computational code underlying the findings in their manuscript fully available?**

Reviewer #1: Yes

Reviewer #2: Yes

PLOS authors have the option to publish the peer review history of their article (what does this mean?). If published, this will include your full peer review and any attached files.

Reviewer #1: No

Reviewer #2: No

---

## [Editor Report · Decision Letter 2]

22 Nov 2024

Dear Dr. Jeanson,

We are pleased to inform you that your manuscript 'Randomness as a driver of inactivity in social groups' has been provisionally accepted for publication in PLOS Computational Biology.

Best regards,

Stuart Johnston

Academic Editor

PLOS Computational Biology

Tobias Bollenbach

Section Editor

PLOS Computational Biology

Feilim Mac Gabhann

Editor-in-Chief

PLOS Computational Biology

Jason Papin

Editor-in-Chief

PLOS Computational Biology

---

## [Editor Report · Acceptance letter]

28 Nov 2024

PCOMPBIOL-D-24-00988R2 

Randomness as a driver of inactivity in social groups

Dear Dr Jeanson,

I am pleased to inform you that your manuscript has been formally accepted for publication in PLOS Computational Biology. Your manuscript is now with our production department and you will be notified of the publication date in due course.

With kind regards,

Zsofia Freund
